# Therapeutic Targeting of Cancer-Associated Fibroblasts in the Non-Small Cell Lung Cancer Tumor Microenvironment

**DOI:** 10.3390/cancers15020335

**Published:** 2023-01-04

**Authors:** Yasushi Shintani, Toru Kimura, Soichiro Funaki, Naoko Ose, Takashi Kanou, Eriko Fukui

**Affiliations:** Department of General Thoracic Surgery, Graduate School of Medicine, Osaka University, 2-2-L5, Yamadaoka, Suita, Osaka 565-0871, Japan

**Keywords:** non-small cell lung cancer, tumor microenvironment, cancer-associated fibroblasts, cancer progression, anticancer resistance, heterogeneity, target therapy, novel markers

## Abstract

**Simple Summary:**

In the tumor microenvironment, cancer-associated fibroblasts (CAFs) have multiple tumor-promoting functions in drug resistance, regulation of the niche of cancer stem cells and formation of the immunosuppressive network. Multiple mechanisms are involved, including production of growth factors, cytokines, and chemokines, as well as extracellular matrix remodeling. While cancer treatment generally targets cancer cells, elucidation of the tumor microenvironment has allowed for successful targeting of cells other than cancer cells involved in the construction of the tumor microenvironment. Cancer cells are prone to develop drug resistance, whereas resistance to treatments targeting the tumor microenvironment may be less likely to develop. Therefore, various treatment strategies against CAFs have attracted attention as possible novel cancer treatments.

**Abstract:**

Lung cancer is the most frequently diagnosed cancer and the leading cause of cancer death worldwide. The most common lung cancer is non-small cell lung cancer (NSCLC), with an overall 5-year survival rate of around 20% because NSCLC is a metastatic disease. A better understanding of the mechanism underlying lung cancer metastasis is therefore urgently needed. The tumor microenvironment involves different types of stromal cells and functions as key components in the progression of NSCLC. Through epithelial–mesenchymal transition (EMT), in which epithelial cells lose their polarity and acquire mesenchymal potential, cancer cells acquire metastatic abilities, as well as cancer stem-cell-like potential. We previously reported that cancer-associated fibroblasts (CAFs) interact with lung cancer cells to allow for the acquisition of malignancy and treatment resistance by paracrine loops via EMT signals in the tumor microenvironment. Furthermore, CAFs regulate the cytotoxic activity of immune cells via various cytokines and chemokines, creating a microenvironment of immune tolerance. Regulation of CAFs can therefore affect immune responses. Recent research has shown several roles of CAFs in NSCLC tumorigenesis, owing to their heterogeneity, so molecular markers of CAFs should be elucidated to better classify tumor-promoting subtypes and facilitate the establishment of CAF-specific targeted therapies. CAF-targeted cancer treatments may suppress EMT and regulate the niche of cancer stem cells and the immunosuppressive network and thus may prove useful for NSCLC treatment through multiple mechanisms.

## 1. Introduction

Survival rates for non-small cell lung cancer (NSCLC) patients after diagnosis have improved substantially in this century, owing to the implementation of chemotherapeutic regimens, novel molecularly targeted therapies, and immunotherapies; however, mortality rates remain high [1]. The main reason for this problem is that NSCLC is a highly metastatic disease, and curative systemic therapies remain lacking. Furthermore, the occurrence of drug resistance represents a major obstacle to successful treatment for NSCLC patients that requires urgent attention [2,3].

The tumor microenvironment comprises different types of stromal cells, including vascular lymphatic networks, fibroblastic cells, inflammatory immune cells, mesenchymal stem cells (MSCs), and extracellular matrix (ECM). These represent key components in tumor progression, and stromal components have been speculated to be functionally organized to promote cancer cell survival, promoting aggressive neoplastic phenotypes and the acquisition of drug resistance [4,5]. A specific subset of stromal cells, termed cancer-associated fibroblasts (CAFs), constitutes a major stromal component [6]. CAFs were initially described as myofibroblasts that are distinct from normal fibroblasts because of their expression of α-smooth muscle actin (α-SMA) and are also shown to synthesize important amounts of collagen and other ECM components [7]. Recently, CAFs have been speculated to have important functions in epithelial solid tumor biology, such as neoplastic progression, tumor growth, and metastasis, via secretion of factors that promote tumorigenesis by stimulating angiogenesis, cancer cell proliferation, and invasion, in addition to recruiting macrophages and suppressing T-cell antitumor immunity [6,8]. CAFs are genetically stable and influence tumor progression and sensitivity to antitumor therapy through the secretion of ECM, cytokines, and growth factors. Recently, many reports have shown that CAFs can modulate the immune system via paracrine signaling, juxtacrine interactions, and ECM remodeling [9,10,11]. In NSCLC, CAFs have been implicated in enhanced stemness; migration; tumor growth; and resistance to chemotherapy, radiotherapy, tyrosine kinase inhibitors (TKIs), and immunotherapy [12,13,14,15]. Therapeutic targeting of CAFs in the NSCLC tumor microenvironment thus represents an intriguing possibility for improving outcomes in NSCLC patients.

Before antitumor roles for CAFs can be proposed, markers to allow for the definition of CAF subsets need to be identified [16,17]. Two commonly used molecular markers are α-SMA and fibroblast activation protein (FAP); however, these markers do not allow for precise detection of CAFs. CAFs are reported to account for 50–70% of all cells in the tumor microenvironment of breast cancer [18]. While they are also most prominent in the lung cancer tumor microenvironment [19], it is unclear what percentage of stromal cells consists of CAFs because of a lack of specific markers. In addition, not all CAFs demonstrate these features. Several studies have presented details showing the wide heterogeneity among CAFs, probably due to multiple types of originating cells of origin, suggesting that specific CAF populations could be exploited as therapeutic targets [20,21,22].

CAFs are negative for epithelial, endothelial, and leukocyte markers and show an elongated morphology and a lack of mutations in cancer cells. Although CAFs are known as a type of activated fibroblast in the tumor microenvironment, they remain poorly defined due to their heterogeneity and their lack of highly specific markers. This review provides an overview of the current knowledge base regarding the roles of CAFs in NSCLC and the implications of these findings for NSCLC treatment.

## 2. CAF Biomarkers and Heterogeneity in NSCLC

### 2.1. Biomarkers for CAFs in NSCLC

Fibroblasts are easily digested and cultured in plastic flasks, whereas other types of cells are not [23]. We defined fibroblasts obtained from NSCLC tumors as CAFs and those from normal lung tissue as lung normal fibroblasts (LNFs) [24]. A passage number from 1 to 6 is suitable for cultured CAFs in experiments to keep their characteristics. Primary CAFs should be negative for epithelial cell adhesion molecule (EpCAM), cluster of differentiation (CD) 31, and CD45. In practice, traditional CAF biomarkers are typically combined with lineage exclusion to identify CAFs [9]. Vimentin is a marker of quiescent CAFs, whereas α-SMA, fibroblast-specific protein 1 (FSP-1), FAP, platelet-derived growth factor (PDGF) receptor (PDGFR)-α, PDGFR-β, tenascin-C, periostin, podoplanin, CD90 (known as Thy-1), integrin β1, caveolin-1, adipocyte enhancer-binding protein, and endoglin are markers of activated CAFs [7,25,26,27] (Table 1).

Each protein has a biological function. Although α-SMA expression is the hallmark of mature myofibroblasts and implicated in contraction and remodeling of the extracellular matrix (ECM), its role in fibroblast function remains poorly understood [28]. FSP1 belongs to the S100 superfamily, also identified as S100A4, and has been implicated in microtubule dynamics, cytoskeletal membrane interactions, calcium signal transduction, and cell cycle regulation, as well as cell growth and differentiation [29,30]. It is also associated with tissue fibrosis; therefore, it is a useful marker for fibroblasts in pulmonary fibrosis [31]. While the initial description of a cell surface antigen notes its expression by reactive stromal fibroblasts associated with epithelial cancer, it was found to be not expressed by normal fibroblasts and therefore termed “fibroblast activation” protein (FAP). This is a type II transmembrane serine protease that shows enzymatic activity, and research conducted with a catalytically mutant FAP has suggested that FAP can have a functional impact independent of its enzymatic activity [32]. As described above, each expressed protein has a complex biological function. Furthermore, each marker defines a different cell population with partial overlapping, although populations also show distinct expression profiles. As a result, there is no definitive single marker that can be used to identify CAFs.

CAFs have different markers and functions according to their heterogeneity [33]. Irvine et al. performed a meta-analysis of all published markers of CAFs in NSCLC to characterize their heterogeneity [34]. They found that five proteins, namely podoplanin, carbonic anhydrase IX, α-SMA, periostin, and FAP, were suitable for meta-analysis and showed that CAF expression of podoplanin or α-SMA was consistently associated with poor prognosis in NSCLC patients. They also concluded that studies linking CAF protein markers to the cellular processes crucial to CAF function are critical to understanding the biology of CAFs.

### 2.2. Heterogeneity of CAFs in NSCLC

Evidence increasingly suggests wide heterogeneity among CAFs due to varying stages of differentiation from cells that have diverse origins such as tissue-resident fibroblasts, endothelial cells, pericytes, adipocytes, and bone-marrow-derived MSCs, as well as epithelial cells and cancer cells [35,36,37]. Recent single-cell RNA sequencing (scRNA-seq) studies of NSCLC have suggested that CAFs represent a collection of cells with diverse molecular features [38,39,40,41,42]. Studies using scRNA-seq have demonstrated various molecular phenotypes among CAFs and have led to the development of new strategies for investigating CAF biology (Table 2). Lambrechts et al. analyzed stromal cells derived from resected NSCLC tumor tissues and non-tumor lung tissues via scRNA-seq, identifying seven molecular subpopulations of fibroblasts [40]. They proposed limited marker genes for each subpopulation, resulting in the identification of five types of fibroblasts in cancer tissues [40]. Whereas one of these five clusters showed an epithelial–mesenchymal transition (EMT) signal in line with the expression of transforming growth factor (TGF)-β-associated genes, one of the other phenotypes expressed *ACTA2* (the gene for α-SMA) with high expression of other genes involved in myogenesis and angiogenesis. Two of the five clusters were highly similar, with lower myogenesis and signature high mechanistic target of rapamycin (mTOR) expression, but expression of glycolysis-related genes, as well as the location in the tumor differed between them. Hu et al. identified three functional subtypes in NSCLC according to hepatocyte growth factor (HGF) and fibroblast growth factor (FGF) 7 expression using a living biobank of CAFs from NSCLC patients [43]. While subtype I and II CAFs have high HGF and FGF7 expressions and protect cancer cells by activating tyrosine kinase receptor-mediated signaling, subtype III CAFs are associated with better clinical response and immune cell migration via the production of chemoattractants for immune cells with enhanced migration of tumor-infiltrating lymphocytes. These functional differences among CAFs are regulated by TGF-β signaling, which suppresses HGF and FGF7 expression. Kim et al. identified subpopulations of CAFs isolated from human lung adenocarcinomas via scRNA-seq cell trajectory analysis and revealed two branch points, allowing the definition of the following subpopulations of lung CAFs: immunosuppressive, neoantigen-presenting, myofibroblastic, and proliferative CAFs [44]. They also showed that expression of karyopherin subunit alpha 2 on CAFs, as a protein involved in the nucleocytoplasmic transport pathway for a variety of tumor-associated proteins and representing one of the neoantigen-presenting CAF-specific markers, influenced tumorigenesis and metastasis of lung cancer cells in a murine model, suggesting that CAF subtype markers may offer therapeutic targets in the tumor microenvironment. Su and colleagues found that CD10 + GPR77 + CAFs promote tumor formation and chemoresistance by providing a niche for cancer stem cells through activation of nuclear factor kappa-light-chain-enhancer of activated B cells (NF-κB) [45].

CAFs should therefore be investigated based on the notion that some CAF subtypes promote tumor growth and others suppress tumor growth in order to clarify novel populations of CAFs relevant as therapeutic targets in NSCLC.

## 3. CAF-Related Signaling Pathways in NSCLC

### 3.1. Signaling Pathways between CAFs and NSCLC

Cancer initiates fibroblast transition with acquisition of CAF phenotypes via cancer-derived growth factors, as well as cytokines that regulate the TGF-β and NF-κB signaling pathways [46]. We have previously shown that TGF-β secreted from NSCLC cancer cells activates fibroblasts in the tumor microenvironment [15]. CAFs secrete growth factors, including TGF-β, FGF2/7, PDGF, and HGF, as well as vascular endothelial growth factor (VEGF), which promote cancer cell proliferation [47]. Resident fibroblasts are also activated and change to proinflammatory CAFs during the early preneoplastic stages of tumorigenesis [6]. Activated CAFs produce higher amounts of stromal cell-derived factor 1 (SDF-1) than LNFs, and SDF-1 facilitates cancer cell proliferation and chemoresistance via the CXC chemokine receptor 4 (CXCR4)-mediated signaling pathway, which involves NF-κB and B-cell lymphoma-extra large [48]. Mitogen-activated protein kinase, phosphatidylinositol 3-kinase (PI3K)/mTOR, and Wnt/β-catenin signaling are also activated in cancer cells in response to CAF-derived growth factors and cytokines. The Janus kinase (JAK)/signal transducer and activator of transcription (STAT) pathway are activated by CAF-derived interleukin (IL)-6, IL-10, IL-11, and IL-22 [10]. We previously reported that NSCLC cells underwent EMT and acquired stem-cell-like properties when cocultured with CAFs isolated from surgical exploration [24]. We also found that IL-6 from CAFs enhanced EMT and chemoresistance in NSCLC cells, suggesting a role of IL-6 in the maintenance of a paracrine loop that functions as part of the communication between CAFs and NSCLC cells [15,49]. Similarly, CAF-derived C-X-C motif chemokine ligand (CXCL)1, IL-6, and cyclooxygenase-2, known targets of the NF-κB transcription factor, reportedly correlate with tumor-promoting inflammation and tumor invasiveness [50,51]. Iwai et al. reported altered gene expression in NSCLC CAFs compared to LNFs using capped analysis of gene expression to comprehensively analyze promoter activity in CAFs [52]. Among 390 genes highly expressed in NSCLC CAFs, they identified collagen type XI α1, integrin α11, and collagen type I α1 as CAF-specific genes that promote CAFs migration toward collagen type I and fibronectin via extracellular signal-regulated kinase 1/2. Thus, several different signaling pathways in CAF-mediated cancer progression have been extensively explored to determine their roles in the tumor microenvironment (Figure 1). All these signaling pathways in CAFs have potential as targets for blocking crosstalk between CAFs and cancer cells.

### 3.2. Role of CAFs in Resistance to Antitumor Therapy

We previously demonstrated that EMT resulted in increased malignant potential and reduced sensitivity to chemotherapy in NSCLC cells [53]. Furthermore, chronic exposure to anticancer drugs or radiation resulted in cells forming therapy-resistant sublines that underwent EMT via interactions between NSCLC cells and CAFs through the TGF-β, and IL-6 pathways [15]. Exposure to anticancer drugs enhances IL-11 secretion by CAFs, which promotes chemoresistance of cancer cells through the STAT3 signaling pathway [51,54]. We also demonstrated that CAF expression of growth arrest-specific 6 (Gas6), a natural ligand of tumor-associated macrophage (TAM) receptors with high affinity for the receptor tyrosine kinase Axl, increases during chemotherapy and promotes proliferation and migration of NSCLC cells [55]. Tumoral Axl activation induces EMT and promotes cell survival and chemoresistance [56]. In addition, CAFs enhanced chemoresistance by repression of caspase-3 and caspase-8 through the activation of the annexin A3/JNK pathway [57]. CAFs also induce acquired chemoresistance through the insulin-like growth factor (IGF) 2/IGF receptor (IGFR)-1 paracrine pathway, which activates IGF2/AKT/Sox2/ATP-binding cassette B1 signaling and upregulation of P-glycoprotein expression in NSCLC cells [58]. Such data suggest that CAFs, in concert with tumor cells and other components of the tumor microenvironment, abet resistance to treatment [59].

### 3.3. Role of CAFs in Oncogene Addicted NSCLC

Numerous driver mutations have been identified in NSCLC, and oncogene addiction provides rationale for molecular targeted therapy [60]. Pellinen et al. recently explored associations of EGFR mutations and CAF subtypes using multiplex fluorescence immunohistochemistry; their findings indicated that gene alterations may affect the properties of CAFs in the tumor microenvironment [61]. That microenvironment provides sustained resistance to molecular targeted therapy. CAFs have a critical role in the resistance of lung cancer to epidermal growth factor receptor (EGFR) TKIs through the induction of EMT via CAF-mediated signaling pathways [62,63]. Yoshida et al. also found that lung adenocarcinoma cell lines became more resistant to EGFR TKIs when cocultured with CAFs expressing podoplanin, suggesting that podoplanin-positive CAFs may be useful for predicting response to EGFR TKIs [14]. CAF-released IL-6 mediates NSCLC acquired resistance to EGFR TKIs through the JAK1/STAT3 pathway [64]. In addition, MET activation is an important mechanism for acquisition of resistance to TKIs [65]. Treatment of EGFR- or MET-addicted NSCLC cancer cells was shown to cause a metabolic shift toward increased glycolysis and lactate production, which develop chemoresistance in NSCLC cells [66]. Another study demonstrated that CAFs increase the expression and phosphorylation of annexin A2 by secretion of HGF and IGF-1, which regulate EMT and EGFR TKI resistance in a paracrine manner [63]. Together, these findings suggest that cotreatment targeting CAFs may further improve the antitumor efficacy of molecular targeted therapy.

### 3.4. Role of Extracellular Vesicles in Communication between CAFs and NSCLC

Exosomes carry and transfer a variety of cargo, including small non-coding RNAs, also known as microRNAs (miRNAs), and long non-coding RNAs (lncRNAs), which have essential roles in cellular communication. Shen et al. compared miRNA expression profiles between CAFs from NSCLC and LNFs from matched healthy lungs and showed downregulation of miR-1 and miR-206 and upregulation of miR-31 in CAFs [67]. CAF-derived miRNAs also contribute to the transformation of LNFs into CAFs [68]. Furthermore, miR-210 in the exosome secreted by CAFs was taken up by NSCLC cells and promoted EMT by targeting upstream frameshift 1, a key factor in a variety of RNA decay pathways, and activating the phosphatase and tensin homolog (PTEN)/PI3K pathway in cancer cells, thereby promoting NSCLC migration and invasion [69]. CAF-derived exosomes also exhibited miR-20a upregulation and promoted NSCLC cell proliferation and chemoresistance via PTEN downregulation following activation of the PI3K/AKT pathway [70]. Cancer cell-derived TGF-β1 activates miR-21 expression in LNFs and induces differentiation to CAFs, which promote the proliferation of cancer cells through the secretion of calumenin [71]. CAF-specific miR-196a promotes NSCLC progression via C-C motif chemokine ligand 2 (CCL2) secretion by directly targeting *ANXA1* (the gene for annexin-A1) which has anti-inflammatory properties [72]. LncRNAs also participate in activation of LNFs to CAFs, whereas activated CAFs can change gene expression and secretion characteristics of NSCLC cells through lncRNAs [73]. Liu et al. screened fibroblast-specific lncRNAs using RAN-seq data and identified LINC01614 promoting the secretion of IL-6 from cancer cells that upregulates LINC01614 in CAFs, constituting a feedforward loop between CAFs and NSCLC cells [74].

Together, the dysregulation of exosomal miRNA and lncRNAs is involved in the dynamic crosstalk between CAFs and NSCLC cells.

## 4. Role of CAFs in the Immune Environment in NSCLC

### 4.1. Impact of CAFs on Immunosuppressive Activity in NSCLC

CAFs represent a major source of immunosuppressive activity in the tumor microenvironment [75]. Immune regulation by fibroblasts originally attracted attention in other fields of research, particularly the field of autoimmune diseases [27]. CAFs have also been shown to exert strong immunomodulatory function in the tumor microenvironment [76]. Nazareth et al. established culture of stromal cells from primary NSCLCs and revealed that CAFs provide multiple complex regulatory signals, which suppress tumor-associated T-cell functions [77].

Fibroblasts recruit bone-marrow-derived monocytes and dendritic cell precursors via secretion of the CCL2/monocyte chemoattractant protein 1 [78,79]. Xiang et al. showed that lung squamous cell carcinoma patient-derived CAFs promoted recruitment of C-C chemokine receptor 2 (CCR2) + monocytes via CCL2 [80]. Interactions between CCL2/CCR2 recruit immunosuppressive cells such as myeloid-derived suppressor cells (MDSCs) and metastasis-promoting monocytes, resulting in the suppression of autologous CD8+ T-cell proliferation and interferon (IFN)-γ production. Inhibition of CAF function reduced the infiltration of immune suppressor cells into tumor stroma. CAFs overexpressing immunoregulatory cytokines such as TGF-β and VEGF may play important roles in the induction of forkhead box P3 (FOXP3) regulatory T cells (Tregs), and the coexistence of Tregs with CAFs correlate with poor outcomes for NSCLC [81]. Infiltration of immune suppressor cell types, including Tregs and MDSCs, in lung cancer stroma was effectively decreased through reductions in SDF-1, prostaglandin (PG)E2, and TGF-β via inhibition of CAF function in vivo [82]. CAFs can directly enhance the differentiation of Tregs through the production of TGF-β, PGE2 and IL-6 [83]. Furthermore, CAF-conditioned medium induced the expression of programmed death-ligand 1 (PD-L1) in NSCLC cells through CXCL2 secretion [84]. Some CAF subsets can directly deactivate the immune system via the expression of PD-L1 to reduce the activation of T cells, and PD-L1 expression on CAFs represents an independent prognostic factor [85]. CAFs cross-present antigens complexed with major histocompatibility complex class I to antigen-specific CD8+ T cells to directly contribute to the suppression of antitumor T-cell responses in an antigen-specific, antigen-dependent manner via PD-L2 and CD95 (known as FAS) ligand engagement, suggesting that CAFs play a role in immunosuppressive activity within the tumor microenvironment by a mechanism dependent on immune checkpoint activation [86].

### 4.2. ECM Production and CAF Barrier Function in NSCLC

ECM dysregulation is associated with the presence of CAFs and the activity of TGF-β and is linked to immunosuppression [87]. Activated fibroblasts modify the composition of ECM by increasing the deposition of new matrix components such as type I collagen and by modulating the expression of matrix metalloproteinases [88]. CAFs impose a physical barrier around the tumor and inhibit the ability of T cells to reach the tumor stroma [89]. Whereas loose fibronectin and collagen regions enhance active T-cell motility in NSCLC, T cells migrate poorly through areas of dense ECM. This was clarified in a real-time imaging study [90]. The resulting immune-excluding phenotype could be counteracted by remodeling the stromal ECM through treatment with collagenase D. A transgenic mouse in which FAP-expressing cells can be ablated showed decreases in both ECM production and T-cell retention in the stroma of Lewis lung carcinomas by a process involving IFN-γ and tumor necrosis factor-α [91]. In this manner, CAFs contribute to the formation of an immunosuppressed microenvironment through the production and remodeling of ECM components.

## 5. Tumor-Suppressing CAF Phenotypes

CAFs facilitate tumorigenesis and cancer development and have also been shown to be involved in tumor suppression [92]. With varying degrees of activation, CAFs are produced with tumor suppression characteristics and are considered to be more prevalent in early-stage cancer, indicating that greater numbers of tumor-promoting CAFs should be involved in advanced-stage cancer. The underlying mechanisms that lead to tumor promotion or tumor inhibition effects of CAFs are largely unknown [93].

Results obtained with transgenic mouse models with depletion of CAFs in the tumor stroma showed that depletion causes rapid tumor progression rather than tumor suppression, providing direct evidence of the protective effects of CAFs against cancer by secretion of tumor-suppressive cytokines, chemokines, growth factors, and miRNAs [94]. In addition, Rix et al. described a mechanism of paracrine effects for NSCLC cells via specific lung CAF populations through secretion of pro- and antitumor proteins such as IGF1/2 and IGF-binding proteins, respectively [95]. The depletion of CAFs accelerates tumor progression, raising caution regarding nonspecific CAF-depletion therapies. CAFs thus play immunoactivating or immunosuppressive roles in the tumor microenvironment due to their heterogeneity, similar to the direct effects on lung cancer cells.

## 6. CAF-Mediated Anticancer Therapies

### 6.1. Targeting CAFs

Genetic deletion and pharmacological inhibition of FAP inhibited tumor growth in an endogenous mouse model of lung cancer driven by the K-rasG12D mutant [96]. High stromal expression of FAP was identified as an independent marker of poor prognosis in patients with NSCLC [97]. FAP+ cells can both promote tumor progression directly and present a barrier to immunotherapies through the production of ECM and direct signaling pathways [75]. As a result, FAP has been targeted in tumors for imaging and therapies using several approaches, including antibodies, FAP inhibitors, vaccines, and chimeric antigen receptor (CAR) T cells [59]. Targeting FAP in human cancer patients with the humanized monoclonal antibody sibrotuzumab, which shows high affinity for binding to FAP [98], or the FAP enzyme inhibitor talabostat [99], which has not demonstrated clinical efficacy in NSCLC patients. This is probably because binding those drugs to tumor cells in the stroma did not cause sufficient drug-dependent cell cytolysis. A DNA vaccine directed against FAP suppressed tumor proliferation by stimulating a CD8+ T-cell-mediated immune response [100]. Duperret et al. presented a DNA vaccine targeting FAP that drives both CD4 and CD8 T cells [101]. Mice vaccinated with recombinant adenoviral vectors containing FAP cDNA produced FAP-specific cytotoxic T lymphocytes capable of destroying FAP-expressing CAFs, suggesting FAP as a potential target for elimination of CAFs that may develop immunogenic tumor vaccines [102]. CARs are genetically engineered to express CAR molecules targeting tumor antigens and recognize and kill tumor cells. FAP-specific CAR T cells were generated and evaluated in vitro and in vivo, showing that FAP-targeted CAR T cells inhibited the proliferation of NSCLC cells by eliminating FAP-positive CAFs [103,104]. While FAP-targeted CAR-T cells caused severe side effects such as bone marrow toxicity and cachexia [105], in a phase I clinical trial using anti-FAP CAR T-cell therapies for malignant pleural mesothelioma patients, a single infusion of anti-FAP CAR T cells was shown to be safe as long as local administration was used [106]. Further genetic modifications to CAR designs are still necessary to reduce the incidence of treatment-related toxicities [107].

Another CAF-depleting therapeutic strategy depends on surface markers of CAFs. CAF markers offer a potential target for antitumor treatments as CAF-targeted therapies, but such approaches will not exert curative effects against solid tumors. As mentioned above, CAF subsets and phenotypes recently discovered through scRNA-seq have provided glimpses of the possibility of actively targeting these subsets, which represent a minor fraction in the tumor microenvironment [108].

### 6.2. Targeting the Signaling Pathways of CAFs

TGF-β plays an important role in the activation of CAFs and interactions between CAFs and immune cells, indicating TGF-β inhibition therapy as a potentially attractive therapy targeting CAFs [23,109]. We previously reported that CAF mediated secretion of IL-6-induced EMT via enhanced TGF-β signaling in NSCLC, which was attenuated by IL-6 blocking antibody [15]. Song et al. presented the antitumor effect of an IL-6-neutralizing antibody, siltuximab, in mouse xenograft models of NSCLC [110]. In preclinical and phase I and II trials of ALD518, a humanized monoclonal antibody targeting of IL-6 appeared to be well-tolerated and ameliorated NSCLC-related anemia and cachexia by controlling oncogene-associated inflammatory pathways [111]. Tocilizumab, a monoclonal antibody specific to the IL-6 receptor, is expected to control tumor-related symptoms and reduce drug resistance and immune-related adverse events following immune checkpoint inhibitor therapy in NSCLC patients [112]. Thus, IL-6 may present an attractive therapeutic target for the treatment of NSCLC. The FGF/FGFR tyrosine kinase signaling pathway also regulates multiple biological events between CAFs and NSCLC cells, and an FGFR blocker significantly suppressed tumor progression and tumor stroma formation [113]. The chemokine receptor CXCR4 and its ligand, SDF-1 (known as CXCL12), are expressed in a variety of NSCLC and play important roles in the activation and immune suppression of CAFs; thus, the CXCL12/CXCR4 axis may be a target for immune intervention [114]. AMD3100, a specific antagonist of CXCR4, is the most potent small-molecule non-peptide inhibitor of the CXCR4/CXCL12 axis via immune modulation in the tumor microenvironment [115,116,117].

JQ1, a bromodomain and extraterminal domain protein inhibitor, shows therapeutic efficacy against various malignancies. JQ1 attenuated the activation of CAFs via the inhibition of Hedgehog and TGF-β pathways [118]. JQ1 also downregulates glutathione peroxidase 8 (GPX8) expression, which is overexpressed in CAFs and is associated with CAF infiltration in the tumor microenvironment of NSCLC [119]. Furthermore, aberrant expression of histone deacetylases (HDACs) is frequent in human cancers and may be involved in the progression of fibrosis in idiopathic pulmonary fibrosis, as well as NSCLC progression [120]. The HDAC inhibitor CUDC-907 suppressed cancer cell proliferation, collagen production, and ECM deposition and inhibited the migration and invasion properties of fibroblasts and CAFs in an animal NSCLC model [121].

### 6.3. Antifibrotic Therapy to Normalize the Tumor Microenvironment

Fibrosis is a critical feature of the tumor microenvironment of many types of solid tumors. Idiopathic pulmonary fibrosis (IPF) is the most common idiopathic interstitial pulmonary disease and is known to increase the risk of NSCLC; thus, there are multiple common genetic, molecular, and cellular processes that connect lung fibrosis with NSCLC, including fibroblast activation [122]. CAF-targeted treatments designed to block the progression of fibrosis are in development [123]. Pirfenidone is clinically used to treat IPF as an antifibrotic agent [124,125]. We previously showed that pirfenidone inhibited not only fibroblast activity but also crosstalk between NSCLC cells and CAFs by a synergistic potential for inhibiting both IL-6 and TGF-β signaling [126]. Periostin is an ECM protein secreted from CAFs and IPF-activated fibroblasts [127,128]. We reported that periostin plays critical roles in the proliferation of NSCLC cells, and the inhibition of periostin–receptor (integrin β3) interactions attenuated the aggressiveness of NSCLC with IPF [129]. Periostin with CAFs impaired lymphatic endothelial barriers and consequently enhanced metastasis [130]. Periostin has variant C-terminal splicing forms, so pharmacological inhibition of cancer-specific or CAF-specific variants of periostin may effectively suppress tumor expansion and overcome chemoresistance [131]. Nintedanib (BIBF1120) is a small-molecule inhibitor of receptor tyrosine kinases, including FGFR, PDGFR, and VEGFR, and has been approved for treatment of lung adenocarcinoma, as well as IPF [132,133]. Kato et al. reported that nintedanib enhances antitumor immunity by targeting CAFs and thereby improves response to an immune checkpoint blockade [134]. Those findings indicate that antifibrotic drugs may be useful as CAF-targeted agents.

### 6.4. Targeting Immunomodulation of CAFs

Immunotherapy does not work well in some NSCLC patients. One major reason for cancers failing to respond to anti-PD-1/PD-L immune checkpoint therapy is that CD8+ T cells cannot infiltrate into the tumor [135]. CAF-rich tumors exhibit an immunologically cold tumor microenvironment by forming a physical barrier [136,137]. Furthermore, CAFs directly attenuate the activation and proliferation of CD8+ and CD4+ T cells, promote the differentiation of Tregs, and recruit MDSCs for tumor progression and treatment resistance. Altogether, CAFs represent an attractive clinical target for improving response rates to immunotherapy in solid tumors.

### 6.5. New Concepts for Targeted Therapies Using CAFs

Using CAFs as a drug delivery tool is another interesting concept. Leucine-rich-repeat-containing 15 (LRRC15) was induced by TGF-β on activated α-SMA+ CAFs. ABBV-085, a LRRC15 antibody drug conjugate used to deliver the potent antimitotic drug monomethyl auristatin E (MMAE), demonstrated preclinical efficacy against LRRC15 in in vivo models of NSCLC, resulting in localization of the MMAE at high levels in the tumor microenvironment and killing tumor cells by the cell-permeable effects of MMAE [138]. A novel class of CAR T cells was engineered to express an enzyme that activates a systemically administered small-molecule prodrug in situ at a tumor site [139]. CAR T cells against CAFs localize drugs in the TME via this delivery ability and reshape the TME for favorable antitumor responses [140].

### 6.6. Clinical Trials Targeting CAFs in NSCLC

The tumor-promoting functions of CAFs make them promising targets for anticancer therapy, providing attractive emerging therapeutic strategies that can be categorized into blockage of CAF recruitment and activation, depletion of CAF populations, blockage of CAF-mediated signals, and remodeling of the ECM (Figure 2).

However, many proposed strategies have failed to show promising clinical outcomes, probably because of the current incomplete understanding of fibroblast heterogeneity. Several clinical trials targeting CAF-related signals in a variety of malignant diseases are ongoing; recent clinical trials involving NSCLC patients are summarized in Table 3.

## 7. Future Perspectives and Conclusion

CAF-targeted cancer treatments may suppress tumor progression and regulate the niche of cancer stem cells and immunosuppressive networks, providing utility in NSCLC treatment through multiple mechanisms. Key signaling pathways provided by CAFs and CAF-derived factors have great potential for targeted therapy. We have shown that IL-6 contributes to the maintenance of a paracrine loop, which functions as part of the communication between CAFs and NSCLC cells. Anti-IL-6 therapies have been used clinically to treat various diseases; thus, a study to reveal an inhibitor of the IL-6/JAK/STAT3 signaling pathway may be valuable for investigations of targeted therapy [15,141,142]. As mentioned before, FAP is preferentially expressed by CAF and is one of the biomarkers used for their identification; thus, a method for targeting FAP was developed to overcome challenges in the tumor microenvironment [143]. Additionally, FAP inhibitors (FAPI) have recently been developed for positron emission tomography (PET) imaging and radioligand therapy, which will be useful for exploring clinical applications in several types of cancer [144]. FAPI PET findings can also be used to predetermine the candidacy for an FAP-expressing CAF-targeted therapy, as well as evaluation of its efficacy.

However, the tumor-promoting or tumor-suppressive status of CAFs may cancel each other out, depending on the tumor microenvironment; thus, analysis of their lineage is necessary. New research technologies, including scRNA-seq, which allows access to changes among stromal, immune, and cancer cells involved in tumor progression, as multiplex spatial analysis methods, have recently been developed [145]. These technologies can provide valuable insight into the effects of cellular crosstalk dynamics and heterogeneity on cancer patient prognosis and response to various treatments [146].

In conclusion, to develop treatment strategies against CAFs, which have attracted attention as potent novel NSCLC treatments, we should understand the subset of functions of CAFs in the tumor microenvironment and identify targeted CAFs in a more specific manner.

## Figures and Tables

**Figure 1 cancers-15-00335-f001:**
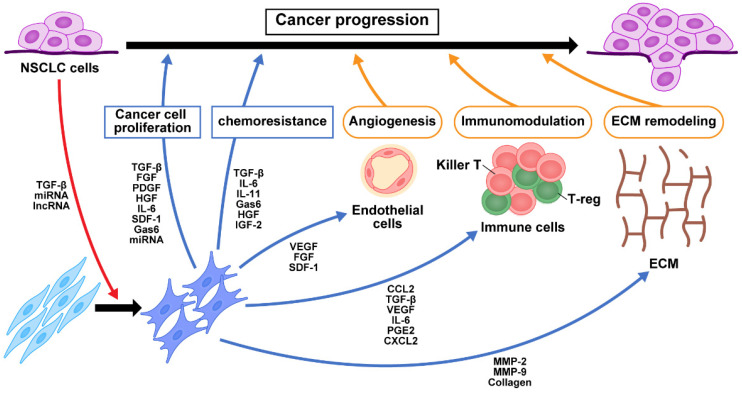
Crosstalk of signaling pathways among CAFs, NSCLC cells, and immune cells. TGF-β and exosomes secreted from NSCLC cancer cells activate fibroblasts into CAFs in the tumor microenvironment. The production of growth factors, cytokines, chemokines, and exosomes from CAFs contribute to NSCLC cell proliferation, chemoresistance, angiogenesis, immunomodulation, and ECM remodeling. These in turn change the tumor microenvironment and contribute to NSCLC progression. NFs, normal fibroblasts; CAFs, cancer-associated fibroblasts; NSCLC, non-small cell lung cancer; miRNA, microRNA; lncRNA, long non-coding RNA; TGF-β, transforming growth factor-β; FGF, fibroblast growth factor; PDGF, platelet-derived growth factor; HGF, hepatocyte growth factor; IL-6, interleukin-6; SDF-1, stromal cell-derived factor-1; Gas6, growth arrest-specific 6; IL-11, interleukin-11; IGF2, insulin-like growth factor 2; VEGF, vascular endothelial growth factor; CCL2, C-C motif chemokine 2; PGE2, prostaglandin E2; CXCL2, chemokine (C-X-C motif) ligand 2; MMP, matrix metalloproteinase; ECM, extracellular matrix.

**Figure 2 cancers-15-00335-f002:**
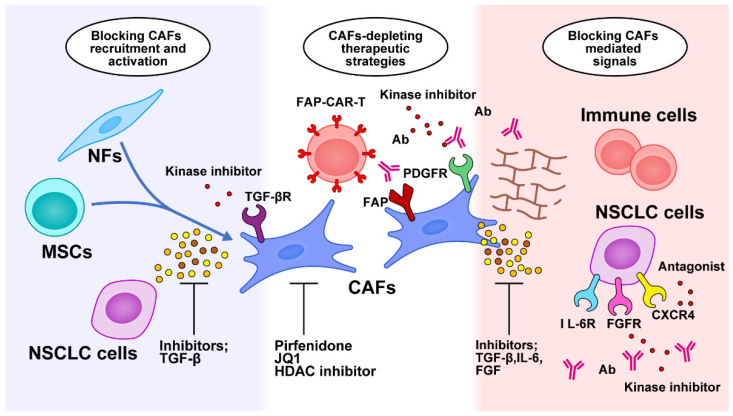
Targeted therapy against CAF-related crosstalk in the tumor microenvironment. Therapeutic strategies are categorized into blocking CAF recruitment and activation, depleting the CAF population, blocking CAF-mediated signals, and remodeling the ECM. NFs, normal fibroblasts; CAFs, cancer-associated fibroblasts; NSCLC, non-small cell lung cancer; MSCs, mesenchymal stem cells; FAP, fibroblast activation protein; CAR-T, chimeric antigen receptor T cell; PDGFR, platelet-derived growth factor receptor; HDAC, histone deacetylase; Ab, antibody; TGF-β, transforming growth factor-β; IL-6, interleukin-6; IL-6R, interleukin-6 receptor; FGFR, fibroblast growth factor; CXCR4, CXC chemokine receptor 4.

**Table 1 cancers-15-00335-t001:** List of biomarkers for CAFs in non-small cell lung cancer.

CAF Marker	Description	Function	Effect on Tumor
Vimentin	Type III intermediate filament protein	Cell structure, integrity	Tumor invasion, metastasis (when expressed in cancer cells)
α-SMA	Actin isoform	Cell structure, integrity, contractility	Tumor proliferation
S100A4/FSP-1	Ca^2+^-dependent S100 family	Cell motility, tissue fibrosis	Tumor invasion, metastasis (when expressed in cancer cells)
FAP	Serine protease	ECM remodeling	Tumor invasion, metastasis
PDGFR-α/β	Tyrosine kinase receptor	Tyrosine kinase activity	Angiogenesis, immunomodulation
Tenascin-C	ECM glycoprotein	Cell proliferation, migration	Angiogenesis
Periostin	ECM protein	Cell proliferation, migration	Tumor proliferation, fibrosis
Podoplanin	Mucin-type transmembrane protein	Cell proliferation, migration	Tumor invasion, metastasis, immunosuppression
Thy-1	Heavily N-glycosylated cell surface protein	Cell–cell interaction	Tumor proliferation, metastasis
Integrin-β1	Transmembrane receptor	Cell–matrix adhesion	Tumor proliferation, metastasis
Caveolin-1	Scaffolding protein within caveolar membranes	Cell signaling, transport	Tumor proliferation, invasion
AEBP	Transcriptional repressor	Gene expression regulation	Tumorigenesis, tumor proliferation
Endoglin	TGF-β co-receptor	Modulation of cellular responses to TGF-β	Angiogenesis

CAFs, cancer-associated fibroblasts; SMA, smooth muscle actin; FSP-1, fibroblast-specific protein 1; FAP, fibroblast activation protein; PDGFR, platelet-derived growth factor receptor; ECM, extracellular matrix; TGF-β, transforming growth factor-β.

**Table 2 cancers-15-00335-t002:** Phenotypic and functional heterogeneity of CAFs in non-small cell lung cancer.

Author	Subtype	Biomarkers	Functions
Lambrechts	Cluster 1	*COL10A1*	EMT signaling
	Cluster 2	*ACTA2*	Myogenesis, angiogenesis
	Cluster 3	-	ECM
	Cluster 4	*PLA2GA2*	Similar to Cluster 1; enriched in leading edge of tumor
	Cluster 5	*MMP3*	Low myogenesis, high mTOR expression
	Cluster 6	*FIGF*	Non-malignant fibroblasts
	Cluster 7	-	Similar to Cluster 4 but differing in glycolysis pathway
Hu	Subtype I	HGF ^High^, FGF ^High/Low^, p-SMAD2 ^Low^	Protects cancer cells, strong TKI rescuer
	Subtype II	HGF ^Low^, FGF ^High^, p-SMAD2 ^Low^	Protects cancer cells, intermediate rescuer
	Subtype III	HGF ^Low^, FGF ^Low^, p-SMAD2 ^High^	Immune cell migration, better clinical response
Kim	Branch 1	*IGFBP6*, *IFITM3*, *LGALS3*	Immunosuppressive CAF, immunomodulation
	Branch 2	*UBE2T*, *TK1*, *CXCL12*, *KPNA2*, *HMGB3*	Neoantigen-presenting CAF, antigen processing and presentation
	Branch 4	-	Myofibroblastic CAF, secretion of cytokines such as CCL2 and TGF-β, leading to fibrosis, immunomodulation, metastasis
	Branch 5	*PRC1*, *AURKA*	Proliferative CAFs, majority of total CAFs
Su		CD10 + GPR77+	Sustaining cancer stemness

COL10A1, collagen type X alpha 1 chain; ACTA2, actin alpha 2, smooth muscle; PLA2GA2, phospholipase A2 group IIA; MMP3, matrix metallopeptidase 3; FIGF, c-fos-induced growth factor; HGF, hepatocyte growth factor; FGF, fibroblast growth factor; IGFBP6, insulin-like growth factor-binding protein 6; IFITM3, interferon-induced transmembrane protein 3; LGALS3, lectin, galactoside-binding, soluble, 3; UBE2T, ubiquitin-conjugating enzyme E2 T; TK1, thymidine kinase 1; CXCL12, C-X-C motif chemokine ligand 12; KPNA2, karyopherin subunit alpha 2; HMGB3, high-mobility group protein B3; PRC1, polycomb repressive complex 1; AURKA, aurora kinase A; CD10, cluster of differentiation 10; GPR77, G protein-coupled receptor 77; EMT, epithelial–mesenchymal transition; ECM, extracellular matrix; mTOR, mammalian target of rapamycin; TKI, tyrosine kinase inhibitor; CAF, cancer-associated fibroblast; CCL2, C-C motif chemokine ligand 2; TGF-β, transforming growth factor-β.

**Table 3 cancers-15-00335-t003:** Clinical trials targeting CAF-related signals in non-small cell lung cancer.

NCT Number	Drug	Mechanism	Title	Status
05386888	GFH018	TGF-βRI kinase inhibitor	A phase 2 Trial of GFH018 in Combination toripalimab given concurrently with platinum-based chemoradiotherapy for participants with unresectable, locally advanced stage III NSCLC	Not yet recruiting
04691817	Tocilizumab	Human anti-IL-6R mAb	A phase Ib-II trial of tocilizumab and atezolizumab in patients with locally advanced or metastatic non-small cell lung cancer refractory to first-line immune-checkpoint-inhibitor-based therapy	Not yet recruiting
04940299	Tocilizumab	Human anti-IL-6R mAb	A phase II study to assess the safety and efficacy of tocilizumab in combination with ipilimumab and nivolumab in patients with advanced melanoma, non-small cell lung cancer, or urothelial carcinoma	Recruiting
03762122	Rogaratinib	FGFR inhibitor	FGFR inhibitor rogaratinib in patients with advanced pretreated squamous cell non-small cell lung cancer (SQCLC) Overexpressing FGFR mRNA. A multicenter, single-arm phase II trial	Active, not recruiting
03827850	Erdafitinib	FGFR inhibitor	A phase II trial to evaluate the efficacy and safety of erdafitinib in patients with advanced squamous NSCLC harboring FGFR genetic alterations after relapse of standard therapy	Recruiting
05210946	Pemigatinib	FGFR inhibitor	A single-arm clinical study of pemigatinib in the treatment of advanced non-small cell lung cancer patients with FGFR gene alterations who had failed standard therapy	Recruiting
04619563	Anlotinib	Multitarget receptor tyrosine kinase inhibitor	A single-arm exploratory clinical study of anlotinib hydrochloride combined with docetaxel in EGFR mutations and advanced non-small cell lung cancer patients who had progressed after targeted therapy and chemotherapy	Recruiting
05460481	Anlotinib	Multitarget receptor tyrosine kinase inhibitor	Anlotinib plus penpulimab in advanced non-small cell lung cancer previously treated with PD-1/PD-L1 inhibitors: a multicenter, single-arm, explorative trial	Recruiting
05465590	MB1707	CXCR4 peptide antagonist	A phase 1 study to evaluate the pharmacokinetics and safety of MB1707 in patients with advanced cancer	Not yet recruiting
01928576	Entinostat	HDAC inhibitor	A phase II Study of epigenetic therapy with azacitidine and entinostat with concurrent nivolumab in subjects with metastatic NSCLC	Active, not recruiting
05141357	HBI-8000	HDAC inhibitor	A phase 2 study to assess the safety and efficacy of hbi-8000 in combination with pembrolizumab for advanced or metastatic NSCLC	Not yet recruiting

CAFs, cancer-associated fibroblasts; NSCLC, non-small cell lung cancer; TGF-βRI, transforming growth factor-β receptor type I; IL-6R, Interleukin-6 receptor; mAb, monoclonal antibody; FGFR, fibroblast growth factor receptor; CXCR4, CXC motif chemokine receptor 4; HDAC, histone deacetylase.

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
