# Peer review of "Therapeutic Targeting of Cancer-Associated Fibroblasts in the Non-Small Cell Lung Cancer Tumor Microenvironment"

_cancers, 2023, doi:10.3390/cancers15020335_

Round 1

Reviewer 1 Report (Previous Reviewer 2)

The authors replied to all my questions and I have no further concerns.

Reviewer 2 Report (Previous Reviewer 4)

I have read the reply which is , in my opinion, exhaustive and the paper can be accepted for publication.

This manuscript is a resubmission of an earlier submission. The following is a list of the peer review reports and author responses from that submission.

Round 1

Reviewer 1 Report

This review entitiled Therapeutic targeting of cancer-associated fibroblasts in the non-small-cell lung cancer tumor microenvironmen, focused on CAF on NSCLC. The authors summarized the progress of CAF and  CAF-targeted cancer treatments and may prove useful for NSCLC treatment through multiple mechanisms. It is useful and systemic review for  NSCLC researh and treatment.

minor revision 

1. Signaling pathways between CAFs and NSCLC . There are many signaling pathways, the authors just introduced all of them, they should discussdeeply some import sigaling way . and it may be together with the part of   Targeting the signaling pathways of CAFs

2. Part of Role of extracellular vesicles in communication between CAFs and NSCLC  should be more content in the manuscript , for this field was research hot in cancer biology

Reviewer 2 Report

Therapeutic targeting of cancer-associated fibroblasts in the non-small-cell lung cancer tumor microenvironment

The review by Shintani and colleagues aims to summarize the importance of cancer-associated fibroblasts (CAFs) in non-small lung cancer. The authors describe the corresponding biomarkers and various phenotypes of CAFs. They also illustrated signaling pathways and the tumor microenvironment through CAFs. They further discuss the current development or potential strategies of CAFs for cancer treatment. They provide comprehensive information in this review but lack some detailed mechanisms and background for each biomarker. The review also required improvement and restructuring of some paragraphs. The following issues must be addressed:

Major comments:

1.     Although the author wrote the definition, background and molecular characteristics of CAF. However, they did not explicitly write about "cancer-associated" fibroblasts. What percentage do they have and what conditions can be called CAF? In addition, in previous studies, some scientists evaluated that CAFs have tumor-promoting and anti-tumor functions. They should do a lot of discussing and provide evidence that they increase readers' income.

2.     I'm glad that the authors have compiled various molecules about CAF in their table. However, they should explain the detailed mechanism of each flag in this section. For example, FAP (fibroblast activation protein) has dual roles in biological processes through enzymatic activity and non-enzymatic functions. FAP relies on endopeptidases and dipeptidases to further catalyze some natural substrates, including collagens, cytokines, and chemokines. I recommend that they clearly describe the background of each person and how they are regulated.

3.     Judging from previously published articles (PMID:34490078), more drugs and strategies have been proposed. Why did the author choose only these few? Is it because of NSCLC or are they just picking out how many clinical trials are going on?

4.     As far as the current conclusion narrative is concerned, future prospects are very vague and unrealistic. They must clearly discuss their point of view and advise in detail which molecular expression or expression level should be used as a cutting point. CAF subpopulations and how they are regulated have been well-known research targets.

5.     They mentioned that “CAFs secrete growth factors, including TGF-β, FGF2/7, PDGF, HGF, and vascular endothelial growth factor (VEGF) to promote cancer cell proliferation [39]. We have previously shown that TGF-β secreted from NSCLC cancer cells activate fibroblasts in the tumor microenvironment and increase their production of interleukin (IL)-6 [13]”. The upper and lower sentences are not equivalent. The first sentence is to explain the importance of the substance secreted by CAF, but the next sentence is the secretion of lung cancer. They need careful revision.

6.     “EGFR- or MET-addicted cancer cells displayed a metabolic shift toward increased glycolysis and lactate production under prolonged treatment with TKIs, leading CAFs to increase HGF production in a NF- B-dependent manner, which activated MET and led to chemoresistance in cancer cells” is difficult to understand. They should revise it.

7.     They discussed CAF in NSCLC. NSCLC contains some typical gene alteration events, including EGFRmut, KRASmut and ALK fusions. Do these events affect the properties or proportion of CAF?

Minor comments:

1.     In this manuscript, the authors only mentioned lncRNAs and did not describe the exact names.

Reviewer 3 Report

This present review by Shintani et al summarizes most of the research work in non-small-cell lung cancer tumor microenvironment, cancer-associated fibroblasts (CAFs) and their contribution in drug resistance, maintaining tumor stem cell niche and other immunosuppressive function. This work could give a new insight in understanding the role of the above factors in tumor cells survival and could help in designing new strategies in tumor treatment. Therefore,   I am  principally supportive of accepting this work for publication. 

Reviewer 4 Report

The paper is overall well writtent and interesting for journal audience. I have some minor suggestions.  it should be clearly exposed that the role of CAF in tumorigenesis cannot be the same (driver) of that plaed by oncogenes activation (addiction). In this perspective I would like to suggest to add a section on the interplay between CAF and addicted tumors. Another point that deserves discussion is the role of CAF in idiopathic pulmonary fibrosis which shares with cancer some biologic programmes, aming wich the activation of myofibroblasts.  A related figure should halp the readers